# A Feasibility Study of a Rapid Lateral Flow Immunoassay Test for Children and Adolescents to Determine SARS-CoV-2 Seroprevalence

**DOI:** 10.3390/v17121553

**Published:** 2025-11-27

**Authors:** Rukiye Saç, Medine Ayşin Taşar, Burcu Ceylan Cura Yayla, Serçin Taşar, Meliha Sevim, Gözde Onay, İlknur Fidancı

**Affiliations:** 1Department of Pediatrics, Ankara Training and Research Hospital, Ankara 06230, Turkey; sercin_gozkaya@yahoo.com (S.T.); melihakantekin@yahoo.com (M.S.); gozdeonay@gmail.com (G.O.); 2Department of Pediatric Emergency, Ankara Training and Research Hospital, Ankara 06230, Turkey; aysintasar@yahoo.com (M.A.T.); drilknurfidanci@gmail.com (İ.F.); 3Department of Pediatric Infectious Diseases, Ankara Training and Research Hospital, Ankara 06230, Turkey; dr.bcc.83@gmail.com

**Keywords:** adolescents, children, SARS-CoV-2, seroprevalence, rapid lateral flow immunoassay

## Abstract

Global seroprevalence of SARS-CoV-2 in children and adolescents has risen to an estimated 56.6%. Limited data on children highlights the need for ongoing surveillance as recommended by the WHO. Understanding this need, in this study we carry out a feasibility study of a rapid lateral flow immunoassay test (LFIAT) and determine SARS-CoV-2 seroprevalence in children and adolescents. The method used in the paper included 202 children (1–18 years old) who were evaluated for SARS-CoV-2 IgG/IgM antibodies using the LFIAT. LFIAT provides results in 20 min. Demographic data was collected through questionnaires. As a result, SARS-CoV-2 antibody seroprevalence was 52.0% (n = 105). No gender differences were observed. Adolescents exhibited higher seroprevalence (63.0%) compared to children aged 1–5.9 (49.3%) and 6–11.9 years old (46.8%), without statistical significance (*p* > 0.05). The median age of seropositive children was 9 years, and 77.1% had a history of PCR-confirmed infection (*p* = 0.001). In conclusion, results are consistent with global data. It was observed that the LFIAT can be easily administered to children and parents had a positive outlook on the test. Findings suggest that LFIAT can be used to assess SARS-CoV-2 immunity, while providing the first seroprevalence estimate for SARS-CoV-2 among healthy children and adolescents in Turkey, utilizing the COVID-19 IgG/IgM Rapid Test-KitA.

## 1. Introduction

On 11 March 2020, the World Health Organization (WHO) declared COVID-19, a global pandemic. By 10 November 2024, over 776.8 million cases of COVID-19 were reported and more than seven million deaths were confirmed to the WHO across 234 countries [1]. However, wastewater surveillance indicates that clinical case detection has likely underestimated the true burden of the disease by a factor of 2 to 19 [2]. In the European Union, wastewater-based epidemiology is being increasingly integrated into public health surveillance frameworks, supported by European Commission initiatives that encourage routine monitoring of SARS-CoV-2 and other pathogens in municipal wastewater [3]. The virus still causes significant health issues, including severe acute illness and post-COVID-19 conditions. The pandemic’s impact varies by country, and the WHO’s capacity to monitor disease spread, severity, and evolution is hindered by reduced surveillance, testing, sequencing, and limited integration into long-term prevention strategies.

Turkey has reported a total of 17,004,677 cases and 101.419 deaths to the WHO. The country experienced five waves of COVID-19. The first in December 2020, followed by the second wave in April 2021, the third in January 2022, the largest fourth wave in February 2022, and the final wave in June 2022 [4]. Although COVID-19 still causes severe illness and death, many countries delayed or stopped reporting data, making it difficult to accurately assess the number of cases and deaths. This reduction in data means that the reported number of cases does not fully reflect effective infection rates [5]. Although new cases and fatalities continue to emerge in neighboring countries, Turkey have not updated COVID-19 data since March 2023 [6].

WHO continues to consider COVID-19 a threat and urges countries to maintain their infrastructure to deal with the virus. Awareness campaigns, societal precautions, surveillance, early clinical care, and vaccination of high-risk groups are recommended [7]. However, there is a knowledge gap regarding SARS-CoV-2 seroprevalence among children. While systematic reviews regarding SARS-CoV-2 seroprevalence in the general population, healthcare workers, and special populations have been conducted, few studies estimate it in children. Naeimi et al., being one of these few studies [8], found that pooled seroprevalence rates increased from 7.39% to 56.69% across pandemic waves. Seroprevalence was higher in lower-middle income countries (21.21%), whereas it was lowest in high-income countries (10.02%).

The seroprevalence of SARS-CoV-2 in healthy children has been investigated in some countries; however, there is limited information about this in Turkey. One of the first seroprevalence studies in Turkey compared SARS-CoV-2 seroprevalence in children with rheumatic diseases and healthy children. Among the 149 subjects included in the study, 19 subjects (12.75%) were positive with an ELISA immunoglobulin test [9]. In another subsequent study conducted in Turkey included 1000 unvaccinated children (0–12 years) who admitted to pediatric departments with various symptoms between October 2020 and April 2022. Being positive for anti-SARS-CoV-2 total Ig consistently increased, 43.4% in 2020, 60% in 2021, and 88.3% in 2022 [10].

The diagnostic performance of various assays for SARS-CoV-2 detection has been crucial during the pandemic. IgM and IgG antibodies are detectable 3–6 days and 8 days after the SARS-CoV-2 infection, respectively. The lateral flow immunoassay (LFIA) is a rapid, simple, and sensitive method for detecting both antibodies in human blood [11]. Rapid diagnostic tests can be offered for surveillance programs and be used as functional tools for fast and inexpensive monitoring of anti-SARS-CoV-2 antibodies in low- or middle-income countries. However, researchers stress the need to validate these kits locally [12]. Compared with traditional laboratory-based assays such as ELISA and CLIA, LFIA offer several practical advantages for point-of-care antibody detection. While ELISA and CLIA are highly sensitive and specific, they require laboratory infrastructure, longer processing times, and trained personnel, which limit their applicability in large-scale or resource-constrained settings. In contrast, LFIA is rapid, inexpensive, and easy to perform, providing results within minutes and allowing the use of capillary whole blood without specialized equipment. These features make LFIA particularly suitable for pediatric populations and community-based seroprevalence studies, especially in regions with limited laboratory capacity [13,14].

This study aims to determine the anti-SARS-CoV-2 IgM and IgG seroprevalence and evaluate the rapid LFIA COVID-19 IgM and IgG test in children, assessing its usability in outpatient clinic settings.

## 2. Materials and Methods

### 2.1. Study Population, Study Design and Setting

A total of 202 children and adolescents aged 1 to 18 years were prospectively enrolled. Participants were recruited from the Children’s Health and Diseases Outpatient Clinic and Children’s Emergency Clinic at the hospital between 1 April 2022, and 31 May 2022. The study population comprised healthy children who had not exhibited fever and respiratory symptoms over the previous month. Individuals who met the definition of a confirmed case of COVID-19 as per the stipulated guideline [15]. Patients were enrolled in the study in accordance with the order in which they presented themselves at the outpatient clinic.

### 2.2. Participants and Eligibility Criteria

After obtaining voluntary consent, children and parents answered the questionnaire, which covered demographic characteristics (age, gender, comorbidities, the family’s and child’s COVID-19 history, and vaccination status). The exclusion criteria includes individuals who had received vaccination, receiving immune suppressive therapy, suffering from immune deficiency, those with a family history of a confirmed case of COVID-19 within the last month, or those exhibiting COVID-19 symptoms. Subjects were divided into three age groups: 1–5.9 years old, 6–11.9 years old, and >12 years old. After completing the questionnaire, each child underwent a rapid LFIA COVID-19 antibody test.

### 2.3. Rapid LFIA Testing Procedure

COVID-19 IgG/IgM Rapid Test Kit is a lateral flow immunochromatographic diagnostic test, designed for the qualitative differential detection of SARS-CoV-2 IgG and IgM antibodies, formed against both the nucleocapsid and receptor-binding domain of the spike protein of SARS-CoV-2 in whole blood, serum, or plasma. LFIA-based tests either detect IgG and IgM antibodies separately, or both antibodies, within a sample. Several companies have produced IgM/IgG detection LFIA kits for diagnosing COVID-19, and these tests have demonstrated high throughput rates [16].

In this study, two test regions coated with anti-human IgM and IgG antibodies are utilized, in conjunction with a control line within the device. 10 µL sample of whole blood, serum, or plasma is sufficient for the test, which can provide results with 97.20% sensitivity and 99.22% specificity [17]. The appearance of the control line indicates a valid test, whereas the color change in the test indicates positive results. A drop of blood is taken from the fingertip under sterile conditions by the researchers, following the instructions for use when administering the test to pediatric patients. The blood sample is mixed with three drops of sample diluent. Results are obtained within approximately 20 min.

### 2.4. Ethical Considerations

The prior authorization of the study was given by the University of Health Sciences, Ankara Research and Training Hospital Ethical Committee (approval number E-22/980). Authors declare, this study was carried out in compliance with relevant laws and guidelines, and with the ethical standards of the Declaration of Helsinki. Any information that could reveal the patient’s identity was excluded, in the manuscript.

### 2.5. Statistical Analysis

The sample size was calculated based on a power analysis conducted using G*Power (Version 3.1.9.6). In the study, reliability was set at 95%, the power at 80%, and the effect size at 0.25. The minimum sample size was calculated to be 53.

SPSS statistics software (version 20) was used for data analysis. The variables were investigated using visual (histograms, probability plots) and analytical methods (Kolmogrov–Simirnov/Shapiro–Wilk’s test) to determine whether they are normally distributed. Median with minimum–maximum values was applied for continuous variables without normal distribution. For the variables without normal distribution, the Mann–Whitney U test compares two independent groups, Kruskal–Wallis test compares more than two independent groups. Numbers and percentages were used for categorical variables. The Pearson Chi-Square and Fisher–Freeman–Halton tests compared the differences in the categorical variables. The value of *p* < 0.05 was statistically significant.

## 3. Results

The study enrolled 202 children and adolescents, with a median age of 8.0 years (1 to 18 years) and a mean age 8.3 ± 4.6 years, of which 111 (55.0%) were female. 9.9% of the total sample had comorbidities (n = 20). The specific diagnoses were asthma (n = 8), anemia (n = 2), neurological diseases (n = 3), nephrological diseases (n = 5), and coeliac disease (n = 2). Table 1 shows clinical and demographic characteristics, stratified by age groups.

The median duration of a previous history of SARS-CoV-2 PCR (+) infection was 4 months (range: 2–20 months), with no statistically significant difference observed across age groups (*p* = 0.803) of the participating children. For family members, the median duration of a prior SARS-CoV-2 PCR (+) infection was 5 months (range: 1–26 months), and similarly, no significant difference was found across age groups (*p* = 0.727). The frequency of children exhibiting symptoms concurrently with their family members’ illness (n = 113) was 37.2% (n = 42). Again, no statistically significant difference was identified across age groups (*p* = 0.564).

Overall, 105 children (52.0%) tested positive for SARS-CoV-2 antibodies using the rapid diagnostic test.

Table 2 shows the demographic and clinical characteristics of children and adolescents based on rapid LFIA test results.

The median age of children with positive LFIA test results was 9 years (range: 1–18) and a mean age 8.8 ± 4.8 years, while those with negative results were younger, with a median age of 7.4 years (range:1–18) and a mean age 7.9 ± 4.4 years. However, no statistically significant difference was observed between the two groups (*p* = 0.181). Children who tested positive for LFIA had a significantly higher rate of prior SARS-CoV-2 PCR (+) infections compared to those with negative results (77.1% vs. 22.9%) (*p* = 0.001). Additionally, children with a family history of COVID-19 were more likely to test positive for LFIA than those without such a history (60% vs. 51.5%), although this difference did not reach statistical significance (*p* = 0.227).

## 4. Discussion

This study aimed to assess the seroprevalence of COVID-19 antibodies in children and, using a rapid IgM-IgG combined LFIA test. The second aim was to evaluate its applicability in the pediatric population. The overall seroprevalence was found to be 52%, with no significant differences between genders. The highest seroprevalence was observed in the adolescent age group (63%), although no statistically significant difference was detected.

Several studies have examined the seroprevalence of SARS-CoV-2 in pediatric populations, and the sociodemographic factors that may be associated with higher rates of infection. Naeimi R et al. [8], stratified the studies by WHO region, stating that pooled seroprevalence rates varied between waves of the pandemic. The rates increased progressively in each wave, from 7.39% to 56.69% [8]. The review also indicated that seroprevalence rates were higher in older children and those living in low-income countries or regions with low human development indices. The highest, pooled seroprevalence was estimated for lower-middle income countries (21.21%, 12.81–31.05%; *I*^2^ = 99.83%), and the lowest seroprevalence for high-income countries (10.02%, 8.46–11.71%; *I*^2^ = 99.63%). The highest seroprevalence was in the South-East Asia region, while the Western Pacific region had the lowest seroprevalence. No gender-based difference was observed regarding the risk of acquiring SARS-CoV-2 infection. Consistent with these findings, our study also revealed no gender-related differences, with the highest seroprevalence observed in the adolescent age group.

In certain countries, the seroprevalence of SARS-CoV-2 in pediatric populations has been intermittently studied. In a seroprevalence study conducted in Germany from January 2020 to June 2022, 59,786 children aged 1–17 years were tested for SARS-CoV-2 antibodies monthly. In November 2021, before the appearance of the omicron variant, the overall seroprevalence was 14.7% (16.2% of school-age children, 13.0% of preschool children). As of June 2022, the overall seroprevalence rate among children aged 1 to 17 years was 73.5%. For children aged 5 to 10 years, the seroprevalence rate was 84.4%, while for those aged 1 to 4 years, it was 66.6%. In the overall collective, seroprevalence increased fivefold from the fall of 2021 to June 2022. The substantial rise in seroprevalence has been attributed to the high percentage of children who have been infected with SARS-CoV-2, in a population that has low vaccination coverage [18]. An increase in infection-induced SARS-CoV-2 seroprevalence across all pediatric age groups was also observed in a national seroprevalence study in the U.S, between September 2021 and February 2022. The seroprevalence rose from 30% to 68% in children aged 1–4 years, from 38% to 77% in children aged 5–11 years, and from 40% to 74% in adolescents aged 12–17 years. The largest increase in each age group was seen during a period corresponding with the surge in cases due to the Omicron variant [19]. The Omicron variant was initially identified in South Africa in November 2021 as a sub-variant of the SARS-CoV-2 virus [20]. In the context of our nation, the initial reports of the Omicron variant emerged in December 2021 [21]. We think that the high rate of seropositivity in children in our study may be since our study was conducted at a time when the Omicron variant was observed in our country.

Another nationwide seroprevalence study conducted in Israel, from January 2020 to March 2021, investigated the under-diagnosis of SARS-CoV-2 infections among children aged 0–15 years. None of the children in the study had received a COVID-19 vaccine. The median age was 8.9 years (interquartile range: 4.1–13.1). The samples were anonymously obtained from individuals who underwent routine or diagnostic blood tests. The prevalence of seropositivity reached 21.8% by March 2021, the final month of the study period. Children from low socio-economic backgrounds had higher odds of being SARS-CoV-2 seropositive compared to those from high socio-economic backgrounds. There were no significant differences in seropositivity based on age group, sex, or ethnicity. This study underscores the importance of regular seroprevalence studies to assess the population’s immunity to SARS-CoV-2. Such studies help identify the proportion of the population that still requires protection and can inform targeted efforts to reach these individuals [22].

The first seroprevalence study in Turkey was conducted between August and October 2020, to assess the asymptomatic seroprevalence of SARS-CoV-2 among pediatric patients with rheumatic diseases and compare to healthy children. Among the 149 subjects included, 19 (12.75%) had positive ELISA immunoglobulin test with no statistically significant difference between children with rheumatic diseases (7.38%) and healthy children (5.36%) (healthy children: 8; juvenile idiopathic arthritis: 5; juvenile systemic lupus erythematosus: 3, familial Mediterranean fever: 3; *p* = 0.644). Antibody positivity was not associated with age, sex, underlying rheumatic conditions, or the treatments received [9].

Another study in our country was conducted between October 2020 and April 2022. It included unvaccinated children under the age of 12, admitted to pediatric outpatient clinics with various symptoms (31% had respiratory symptoms). No antibody positivity was detected in October 2020, by the ELISA method. However, as SARS-CoV-2 variants emerged and cases increased subsequently, serum antibody positivity rates rose to 92.5% by March 2022. 43.4% of children tested positive in 2020, 60% in 2021, and 88.3% in 2022 (*p* < 0.05). Total Ig positivity rates showed no significant difference between girls and boys. A difference in median antibody levels was observed in two different age groups (123.55 Arbitrary Unit (AU)/mL for 0–5 years and 195.75 AU/mL for 6–12 years); however, this was not statistically significant (*p* > 0.05) [10].

In our study group, the history of PCR-positive COVID-19 rate was higher among children who tested positive on LFIA test compared to those who tested negative. Despite a lower percentage of children reporting past PCR-positive COVID-19 history (17.3%, with a median of 4 months prior), the overall seroprevalence remained high. This difference may be attributed to two reasons. First, it was found that a considerable proportion of families (55.9%) had a history of COVID-19 and 37.2% of children showed symptoms simultaneously with family members (median 5 months ago). Secondly, a considerable proportion of children had asymptomatic infection.

In a cross-sectional study conducted in Germany, blood samples were collected from 1131 children, aged 6 months to 17 years during routine blood draws, between December 2020 to August 2022. The seroprevalence rates were 19.1% in children under 5 years of age, 20.5% in children aged 5–11 years, and 32.8% in adolescents aged 12–17 years. Main risk factor despite the time at risk for silent infections was an infected household member Factors associated with overall infections (known and silent) also include the infection of a household member [23].

Previous studies have demonstrated that children and adolescents are also susceptible to SARS-CoV-2 infections like adults but are more likely to be asymptomatic [24]. Forrest et al., reported that, as the total number of COVID-19 cases rises, the absolute number of children with symptomatic and severe diseases, such as multisystem inflammatory syndrome in children (MIS-C), also increases [25]. Childhood MIS-C is a serious, life-threatening COVID-19 associated illness, and fully vaccinated children are protected from both COVID-19 infections and related complications, including MIS-C [26]. However, in Turkey, only children aged over 12 are eligible for vaccination, which restricts vaccine access among younger children. In our country, the Minister of Health announced on 2 September 2021, that the COVID-19 vaccine would be made available voluntarily to people aged 12 and over [27]. Since 2 December 2024, there has been an absence of data pertaining to vaccination [28].

Rapid, accurate, simple, and inexpensive diagnostic tests such as the LFIA, to detect both IgM and IgG against SARS-CoV-2 are advantageous as they provide better patient triage. These tests can help to lessen the spread of outbreaks by fast recognition of infected people. They are suitable for laboratory facilities in regular clinics, without the need for a skilled practitioner, and they have lower test costs [14,29].

In this study, it was observed that the LFIA test can be easily administered in children under outpatient clinic conditions. In addition, parents were willing to have the test performed. These tests provide results in as little as 15–20 min and can be particularly useful in outpatient settings and low economic resources. However, the major limitation of LFIA tests is their relatively lower sensitivity compared to more robust methods like Enzyme-Linked Immunosorbent Assays (ELISA). Nevertheless, LFIA can be used for serological surveillance, for monitoring antibodies after vaccination and in quality control assessments for developed vaccines. However, confirmatory tests are required for clinical decisions [14,29].

One major limitation in our study was the small sample sizes due to the number of LFIA test kits available. An additional limitation is that although the LFIA test provided valuable insights into seroprevalence, it was not compared with other established tests such as ELISA or CLIA. Such comparative testing would have allowed a more comprehensive evaluation of antibody responses and test performance.

Although LFIA is practical and inexpensive, conducting large-scale seroprevalence studies in children may be challenging, especially in countries with limited resources. However, several approaches can make such studies feasible. LFIA testing can be integrated into routine public health activities such as school-based health screenings, primary care visits, or mobile outreach programs, reducing additional operational costs. Training non-specialist health workers to perform and interpret LFIA, and using simplified data collection platforms, may also facilitate wider implementation. Owing to its low infrastructure requirements and rapid turnaround time, LFIA remains a suitable tool for large-scale monitoring in resource-constrained settings when sampling strategies are organized efficiently.

## 5. Conclusions

As a conclusion, in our study, 52% of asymptomatic children tested positive for SARS-CoV-2 antibodies, aligning with global data. This study is one of the few screening studies conducted in a healthy Turkish pediatric cohort. The LFIA test was practical and easy to perform, easing the rapid isolation of children, a key source of adult transmission. Additionally, parents had a positive outlook about the test. These results suggest that LFIA test can be used to assess SARS-CoV-2 immunity in children. We emphasize the need for additional seroprevalence studies to measure the immunity levels of the population, among different age groups, and plan future public health interventions.

## Figures and Tables

**Table 1 viruses-17-01553-t001:** Demographic Characteristics of Children and Adolescents by Age Groups.

	Age Groups n (%)				*p* ^4^
	Group 1 ^1^ n = 71 (35.1)	Group 2 ^2^ n = 77 (38.2)	Group 3 ^3^ n = 54 (26.7)	Total n = 202 (100.0)	
Gender Female	37 (52.1)	41 (53.2)	33 (61.1)	111 (55.0)	0.563
Comorbidity	6 (8.5)	7 (9.1)	7 (13.0)	20 (9.9)	0.673
Number of family members, median (range)	4 (3–11)	4 (3–10)	4 (2–9)	4 (2–11)	0.067
PCR confirmed COVID-19 history	10 (14.1)	10 (13.0)	15 (27.8)	35 (17.3)	0.059
PCR confirmed COVID-19 history in family	46 (64.8)	44 (57.1)	23 (42.6)	113 (55.9)	0.045
LFIA ^5^ test Positive Negative	35 (49.3) 36 (50.7)	36 (46.8) 41 (52.2)	34 (63.0) 20 (37.0)	105 (52.0) 97 (48.0)	0.161

^1^ Group 1: 1–5.9 years old. ^2^ Group 2: 6–11.9 years old. ^3^ Group 3: 12–18 years old. ^4^ *p*-values < 0.05 were considered statistically significant. ^5^ LFIA: Lateral flow immunoassay.

**Table 2 viruses-17-01553-t002:** Comparison of Demographic and Clinical Characteristics of Children and Adolescents Based on Lateral Flow Immunoassay Test Results.

	Rapid LFIA ^1^ Test Result		*p* ^2^
	Positive n = 105 (52.0%)	Negative n = 97 (48.0%)	
Gender, n (%), Female Male	54 (51.4) 51 (48.6)	57 (58.8) 40 (41.2)	0.295
Comorbidities (n = 20), n (%)	12 (11.4)	8 (8.2)	0.449
Age (years), median (range)	9 (1–18)	7.4 (1–18)	0.181
Number of family members, median (range)	4 (2–10)	4 (3–11)	0.923
History of PCR (+) COVID-19 (n = 35), n (%)	27 (77.1)	8 (22.9)	0.001
Median month of past PCR (+) COVID-19 history in the child, median (range)	4 (2–13)	6.5 (3–20)	0.254
History of PCR (+) COVID-19 in the family (n = 113), n (%)	63 (60.0)	50 (51.5)	0.227
Median month of past PCR (+) COVID-19 history in the family, median (range)	5 (1–23)	5 (1–26)	0.878
Did the child experience any symptoms at the same time? Yes (n = 43), n (%)	26 (60.5)	17 (39.5)	0.429

^1^ LFIA: Lateral flow immunoassay. ^2^ *p*-values < 0.05 were considered statistically significant.

## Data Availability

All data generated or analyzed during this study are included in this published article.

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
