# Peer review of "A Feasibility Study of a Rapid Lateral Flow Immunoassay Test for Children and Adolescents to Determine SARS-CoV-2 Seroprevalence"

_viruses, 2025, doi:10.3390/v17121553_

Round 1

Reviewer 1 Report

Comments and Suggestions for Authors

The manuscript describes SARS-CoV-2 seroprevalence among Turkish children.

  1. As the Authors included in the Introduction data on SARS-CoV-2 from wastewater testing (wastewater base epidemiology), in the Reviewer's opinion, referring to the relevant documents, it should also be stated that from 2026, European Union countries will be obliged to monitor municipal wastewater for dangerous pathogens, including the SARS-CoV-2 virus.
  2. Line 48, 55, 68, 69, 72… (please review the entire text from this aspect): Please correct Türkiye to Turkey.
  3. Please divide the “Materials and Methods” section into subsections to make the information clearer for readers.

In the reviewer's opinion, the manuscript does not contain any novelty, but it presents data describing children, a population that is usually the least described in the literature. For this reason, the reviewer decided that, after revisions, the manuscript could be published.

Author Response

Comment 1: “As the Authors included in the Introduction data on SARS-CoV-2 from wastewater testing... it should also be stated that from 2026, European Union countries will be obliged to monitor municipal wastewater for dangerous pathogens including SARS-CoV-2.”

Response 1:

Thank you for this helpful comment. We agree that wastewater-based epidemiology has become an increasingly important component of public health surveillance in the European Union. In line with the reviewer’s suggestion, we have added a sentence to the Introduction highlighting the European Commission’s initiatives that promote routine municipal wastewater monitoring for SARS-CoV-2 and other priority pathogens. We believe this addition strengthens the context for wastewater-based surveillance mentioned in our manuscript. A new sentence has been added to the Introduction (page 2, lines 43–46).

Comment 2: “Line 48, 55, 68, 69, 72… Please correct Türkiye to Turkey. (Please review the entire text from this aspect).”

Response 2:

Thank you for pointing this out. We have carefully reviewed the entire manuscript and corrected all occurrences of “Türkiye” to “Turkey” as suggested. All instances of “Türkiye” have been revised to “Turkey” throughout the text.

Comment 3: “Please divide the ‘Materials and Methods’ section into subsections to make the information clearer for readers.”

Response 3: Thank you for this helpful suggestion. We agree that structuring the Materials and Methods section into subsections improves clarity. Accordingly, we have reorganized this section and divided it into clearly labeled subsections, including Study Population, Study Design and Setting, Participants and Eligibility Criteria, LFIA Testing Procedure, and Statistical Analysis.

Reviewer 2 Report

Comments and Suggestions for Authors

This is a valuable study that has clarified precisely how many children in Turkey possess SARS-CoV-2 antibodies.
As humans become adapted to this virus, this proportion is expected to increase. I believe it is a positive sign that it has reached roughly half.
It should be conducted regularly in various parts of the world, and the fact that LFIA proved useful in this context is also significant.

However, extending LFIA to a larger cohort appears challenging. The authors should discuss potential methodologies for conducting comparable studies in countries with more limited resources.

Author Response

Comment 1:This is a valuable study that has clarified precisely how many children in Turkey possess SARS-CoV-2 antibodies. As humans become adapted to this virus, this proportion is expected to increase. I believe it is a positive sign that it has reached roughly half. It should be conducted regularly in various parts of the world, and the fact that LFIA proved useful in this context is also significant. However, extending LFIA to a larger cohort appears challenging. The authors should discuss potential methodologies for conducting comparable studies in countries with more limited resources.”

Response 1:

Thank you for this valuable comment. We agree that implementing LFIA-based seroprevalence studies on a larger population scale may be more difficult, especially in countries with limited resources. We have expanded the Discussion section to address potential strategies that could facilitate the use of LFIA in broader or resource-constrained settings. The new text highlights alternative sampling strategies, community-based surveillance models, and cost-effective operational approaches that may improve feasibility.

A new paragraph has been added to the Discussion section (page 8, line 311-320).

Reviewer 3 Report

Comments and Suggestions for Authors

I congratulate the authors, the article is interesting and scientifically valid. I think it is very interesting given the high prevalence of SARS-CoV-2 respiratory infections. I have no particular remarks as the work, its organization and the presentation of the results are done in an excellent way.

I have only a few minor observations.

  1. In the Introduction section (Lines 76–78), the authors state that IgM and IgG are detectable 3–6 days to 8 days after SARS-CoV-2 infection. I would suggest the authors add the word "respectively" to the end of the sentence, to specify that IgM is detectable first and IgG only later.
  2. Also in the Introduction section (lines 78–82), the authors provide a brief description of the LFIA (lateral flow immunoassay) method used for the detection of IgM/IgG. I would suggest the authors add a couple of lines of commentary mentioning the advantages of a method like LFIA compared to traditional methods used for the detection of IgG and IgM, such as ELISA (enzyme-linked immunosrbent assay) and CLIA (chemiluminescence immunoassay), indicating the differences and advantages of LFIA. A couple of lines of commentary is sufficient.
  3. In the Results section (lines 137–138), the authors provide the mean age of the 202 subjects enrolled in this study, indicating the maximum and minimum ages in parentheses. I would suggest the authors also provide the standard deviation of the mean age (mean age +/- SD).
  4. In lines 169–170 of the results statement, the authors indicate the mean age of the group of subjects who tested positive for the LFIA test compared to the group with a negative LFIA test. Here, too, I would suggest indicating the corresponding standard deviation next to the mean age.
  5. In the Results section (lines 169–176), the authors describe that there are no statistically significant differences in age distribution between the LFIA-positive group and the LFIA-negative group. They also emphasize that among the positive group, there is a higher rate (77.1%) of subjects with positive SARS-CoV-2 PCR results, compared to the LFIA-negative group (only 22.9% PCR positives). I would ask the authors to indicate whether the two groups of subjects tested (LFIA-positive and LFIA-negative) were tested with any other commercial method for detecting IgG/IgM, such as an ELISA, for comparison. If this analysis has been performed and the data are available, the data should be shown to further strengthen the work.

Author Response

Reviewer 3

Comment 1: “In the Introduction section (Lines 76–78), the authors state that IgM and IgG are detectable 3–6 days to 8 days after SARS-CoV-2 infection. I would suggest the authors add the word ‘respectively’ to specify that IgM appears first and IgG later.”

Response 1:

Thank you for this helpful observation. We agree with the reviewer’s suggestion. The word “respectively” has been added to clarify the sequence in which IgM and IgG antibodies appear following SARS-CoV-2 infection.

Comment  2: “Also in the Introduction section (lines 78–82), the authors provide a brief description of the LFIA (lateral flow immunoassay) method used for the detection of IgM/IgG. I would suggest the authors add a couple of lines of commentary mentioning the advantages of a method like LFIA compared to traditional methods used for the detection of IgG and IgM, such as ELISA (enzyme-linked immunosrbent assay) and CLIA (chemiluminescence immunoassay), indicating the differences and advantages of LFIA. A couple of lines of commentary is sufficient.”

Response 2:

We appreciate this valuable suggestion. A brief explanation highlighting the main advantages of LFIA over ELISA and CLIA has been added to the Introduction. This addition improves clarity and contextual understanding of why LFIA was chosen in this study. A new paragraph has been added to the Introduction section (page 2-3, lines 86-94).

Comment 3: “In the Results section (lines 137–138), the authors provide the mean age of the 202 subjects enrolled in this study, indicating the maximum and minimum ages in parentheses. I would suggest the authors also provide the standard deviation of the mean age (mean age +/- SD).”

Response 3:

Thank you for your comment. The standard deviation for the mean age of the 202 participants has now been added to the Results section (line 153).

Comment 4: “In lines 169–170 of the results statement, the authors indicate the mean age of the group of subjects who tested positive for the LFIA test compared to the group with a negative LFIA test. Here, too, I would suggest indicating the corresponding standard deviation next to the mean age”

Response 4:

Thank you for this helpful suggestion. As recommended, we have added the standard deviation values next to the mean ages of both the LFIA-positive and LFIA-negative group to the Results section (lines 185–186).

Comment 5:In the Results section (lines 169–176), the authors describe that there are no statistically significant differences in age distribution between the LFIA-positive group and the LFIA-negative group. They also emphasize that among the positive group, there is a higher rate (77.1%) of subjects with positive SARS-CoV-2 PCR results, compared to the LFIA-negative group (only 22.9% PCR positives). I would ask the authors to indicate whether the two groups of subjects tested (LFIA-positive and LFIA-negative) were tested with any other commercial method for detecting IgG/IgM, such as an ELISA, for comparison. If this analysis has been performed and the data are available, the data should be shown to further strengthen the work.”

Response 5:

Thank you for this important clarification request. No additional ELISA or CLIA testing was performed in this study; therefore, LFIA results could not be directly compared with other commercial serological assays. The Limitations section has been revised accordingly.